# Potential “Therapeutic” Effects of Tocotrienol-Rich Fraction (TRF) and Carotene “Against” Bleomycin-Induced Pulmonary Fibrosis in Rats via TGF-β/Smad, PI3K/Akt/mTOR and NF-κB Signaling Pathways

**DOI:** 10.3390/nu14051094

**Published:** 2022-03-05

**Authors:** Yifei Lu, Yihan Zhang, Zhenyu Pan, Chao Yang, Lin Chen, Yuanyuan Wang, Dengfeng Xu, Hui Xia, Shaokang Wang, Shiqing Chen, Yoong Jun Hao, Guiju Sun

**Affiliations:** 1Key Laboratory of Environmental Medicine and Engineering of Ministry of Education, Department of Nutrition and Food Hygiene, School of Public Health, Southeast University, Nanjing 210009, China; luyifei3377@163.com (Y.L.); zhangyihan425@163.com (Y.Z.); zhenyupan96@163.com (Z.P.); wenzhengwuguan@yeah.net (C.Y.); friendlin@126.com (L.C.); wyy@seu.edu.cn (Y.W.); withxu@seu.edu.cn (D.X.); huixia@seu.edu.cn (H.X.); shaokangwang@seu.edu.cn (S.W.); 2Palm Oil Research and Technical Service Institute of Malaysian Palm Oil Board, Shanghai 201108, China; chenshiqing@mpob.com.cn (S.C.); jhyoong@mpob.com.cn (Y.J.H.)

**Keywords:** carotene, preventive effects, pulmonary fibrosis, tocotrienol-rich fraction

## Abstract

Background: Pulmonary fibrosis (PF) is a chronic, progressive, and, ultimately, terminal interstitial disease caused by a variety of factors, ranging from genetics, bacterial, and viral infections, to drugs and other influences. Varying degrees of PF and its rapid progress have been widely reported in post-COVID-19 patients and there is consequently an urgent need to develop an appropriate, cost-effective approach for the prevention and management of PF. Aim: The potential “therapeutic” effect of the tocotrienol-rich fraction (TRF) and carotene against bleomycin (BLM)-induced lung fibrosis was investigated in rats via the modulation of TGF-β/Smad, PI3K/Akt/mTOR, and NF-κB signaling pathways. Design/Methods: Lung fibrosis was induced in Sprague-Dawley rats by a single intratracheal BLM (5 mg/kg) injection. These rats were subsequently treated with TRF (50, 100, and 200 mg/kg body wt/day), carotene (10 mg/kg body wt/day), or a combination of TRF (200 mg/kg body wt/day) and carotene (10 mg/kg body wt/day) for 28 days by gavage administration. A group of normal rats was provided with saline as a substitute for BLM as the control. Lung function and biochemical, histopathological, and molecular alterations were studied in the lung tissues. Results: Both the TRF and carotene treatments were found to significantly restore the BLM-induced alterations in anti-inflammatory and antioxidant functions. The treatments appeared to show pneumoprotective effects through the upregulation of antioxidant status, downregulation of MMP-7 and inflammatory cytokine expressions, and reduction in collagen accumulation (hydroxyproline). We demonstrated that TRF and carotene ameliorate BLM-induced lung injuries through the inhibition of apoptosis, the induction of TGF-β1/Smad, PI3K/Akt/mTOR, and NF-κB signaling pathways. Furthermore, the increased expression levels were shown to be significantly and dose-dependently downregulated by TRF (50, 100, and 200 mg/kg body wt/day) treatment in high probability. The histopathological findings further confirmed that the TRF and carotene treatments had significantly attenuated the BLM-induced lung injury in rats. Conclusion: The results of this study clearly indicate the ability of TRF and carotene to restore the antioxidant system and to inhibit proinflammatory cytokines. These findings, thus, revealed the potential of TRF and carotene as preventive candidates for the treatment of PF in the future.

## 1. Introduction

The severe acute respiratory disease caused by severe acute respiratory syndrome coronavirus 2 (SARS-CoV-2), which was first reported from Wuhan, China, in December 2019 [1], has spread rapidly to become a global pandemic, with more than 260 million confirmed infections and almost 5.2 million deaths reported by the World Health Organization (WHO) as of 28 November 2021. Although more than 230 million people worldwide were then registered as having recovered from COVID-19, long-term pulmonary complications remain a concern [2], particularly while the spread of the pandemic continues. Interstitial pneumonia is a common feature of COVID-19 and can be complicated by acute respiratory distress syndrome (ARDS), of which pulmonary fibrosis (PF) is a recognized sequela [3]. Fibrotic changes have been reported in patients experiencing severe or long-term COVID-19, as well as in the pulmonary postmortems of patients with COVID-19 [4]. Within the gravity of the pandemic, the burden of fibrotic lung disease following SARS-CoV-2 infection is considered high, and approximately 47% of COVID-19 patients have been reported to suffer impaired gas transfer consistent with pulmonary fibrosis or associated vasculopathy [5]. Therefore, PF is a subject of growing discussion and concern among medical and scientific communities worldwide.

PF literally means scarring in the lungs and it is the devastating consequence of various inflammatory diseases of the lung [6,7]. Although PF is incurable, early-stage injury and immune disorders are preventable and can be corrected. Therefore, it is particularly important for high-risk and suspected groups to prevent and correct related front-end symptoms in their early stages, to prevent the progressive development of PF [8]. The two drugs mainly used in the treatment of PF, namely pirfenidone and nintedanib, have limited efficacy and tend to cause unwanted side effects, including gastrointestinal problems (dyspepsia and anorexia) and dermatological side effects (photosensitivity) from pirfenidone and different gastrointestinal issues (diarrhea and nausea) from nintedanib [9]. These two drugs also present a financial burden for patients and do not contribute to the prevention of PF, so it is essential that a novel treatment with suitably preventive potential is urgently developed. The most commonly used animal model for clinical experiments in this field is that of bleomycin-induced fibrosis, since it is considered an effective drug-induced lung disease [10], and it was therefore selected as the modeling method for use in this study.

Phytochemicals, which are known natural products produced by plants, have been used in the management of PF in China for several years [11] and there have also been many international studies into their potential benefits for PF patients [6,12]. Research involving the effects of these natural products have targeted inflammatory injury, oxidative stress, fibroblast activation, metabolic regulation, extracellular matrix accumulation, and epithelial–mesenchymal transition (EMT), and have also included the TGF-β1/Smad, p38 mitogen-activated protein kinase (MAPK), PI3K/Akt, Nrf2-Nox4, NF-κB, and adenylate-activated protein kinase (AMPK) signaling pathways [9,13,14,15,16,17]. Although the vitamin family plays an important role in antioxidant and anti-inflammatory functions, its influences on PF have not been fully investigated. Vitamin A (retinol) is an essential nutrient for lung, heart, and liver tissues [18,19]; however, among these, the lung is the most sensitive to the influences of vitamin A [20]. The phytonutrients alpha (α)- and beta (β)-carotene often act as precursors for the synthesis of vitamin A. Carotene also plays a role in the management of oxidative stress, which results from infectious and inflammatory processes [21] and may contribute to the pathogenesis of diffuse lung disease (DLD) [22]. Vitamin E, which is an effective antioxidant, can be grouped into two main categories, namely tocopherols and tocotrienols [22,23,24]. Of the two, tocotrienols have been proven to have significantly more effective anti-inflammatory and antioxidant properties [25]. Since its discovery in 1956, several clinical experiments have proven the safety, efficacy, and tolerability of tocotrienols [26,27], and recent research has revealed its superior anti-cholesterolemic (unique to tocotrienols), cardioprotective, antioxidant, anti-cancer, anti-inflammatory, and neuroprotective properties. Although there has been steady growth in the scientific interest in tocotrienols since 1966, it accounts for only 3% of the published research articles related to vitamin E (VE) listed in PubMed [25]. There are limited studies that suggest that both the molecular and therapeutic targets of the tocotrienols are distinct from those of the tocopherols, such as the suppression of inflammatory transcription factor NF-κB, which is closely linked to PF [22]. Furthermore, pharmacokinetics studies indicate that, despite incomplete absorption, oral administration can provide the highest rate of bioavailability compared to other routes of administration, such as intraperitoneal and intramuscular, in which absorption is negligible [22,28]. The abovementioned findings suggest a potential beneficial link between PF and tocotrienol and carotene interventions, and whether their combined effect can play a more effective role in the prevention of PF, both of which are, therefore, explored in this study.

## 2. Materials and Methods

### 2.1. Chemicals and Reagents

Tocotrienol-rich fraction (TRF) and natural mixed-carotene complex 20% oil concentrate (carotene) samples were obtained from the Malaysian Palm Oil Board (MPOB). The TRF and carotene samples were purified from palm oil through molecular distillation technology. Bleomycin (BLM) was purchased from Thermo Fisher Scientific (Waltham, MA, USA). Interleukin-1β (IL-1β), interleukin-6 (IL-6), myeloperoxidase (MPO), while transforming growth factor beta 1 (TGF-β1) and tumor necrosis factor-alpha (TNFα) ELISA kits were acquired from Nanjing Jin Yibai Biological Technology Co., Ltd. (Nanjing, China). Catalase (CAT), glutathione (GSH), malondialdehyde (MDA), nitric oxide (NO), and superoxide dismutase (SOD) were purchased from Nanjing Jiancheng Technology Co., Ltd. (Nanjing, China). Polyvinylidene fluoride (PVDF) membranes and bicinchoninic acid (BCA) protein concentration assay kits were purchased from Millipore (Massachusetts, USA) and Biyuntian Reagent Co., Ltd. (Shanghai, China), respectively. Antibodies were purchased from Abcam (Shanghai, China) as well as from Proteintech (Waltham, MA, USA).

### 2.2. Animals and Experimental Design

The protocols followed throughout the experiments in this study were approved by the Animal Research Ethics Committee of Southeast University (Animal Ethics Approval No. 20201115010). Male Sprague-Dawley (SD) rats (200–220) g, within 6–7 weeks, specific pathogen-free (SPF) grade, Certification No. 20170005039090) were purchased from Shanghai Laboratory Animals Center (SLAC) (Shanghai, China). The rats were housed in cages at room temperature (23 ± 2 °C) with 60 ± 10% humidity and maintained in a 12:12 h light/dark cycle. The rats were free to have water and forage. After a seven-day observation period, the rats were randomly divided into the following seven groups (*n* = 11): a control group, a model group, a low dose TRF (50 mg/kg body wt/day) group, a medium dose TRF (100 mg/kg body wt/day) group, a high dose TRF (200 mg/kg body wt/day) group, a TRF (200 mg/kg body wt/day) + carotene (10 mg/kg body wt/day) group, and a carotene (10 mg/kg body wt/day) group. A single intratracheal instillation of BLM (5 mg/kg) was carried out to induce PF in the treated SD rats, while those in the control group received an equal volume of normal saline. A single day after BLM induction, the rats in the experimental groups were intragastrically administrated with TRF or carotene dissolved in 0.5% sodium carboxymethyl cellulose (CMC-Na), while the rats in the control and BLM groups were intragastrically administrated with an equal volume of 0.5% CMC-Na for 28 days. Body weight was measured daily to monitor changes in growth. After the 28th day, all rats were euthanized and their corresponding organs were collected for analysis.

### 2.3. Preparation and Examination of Samoles

TRF was weighed accurately (0.01 g) (with a 4 decimal-point balance) into a volumetric flask and the exact weight of the sample was recorded. Then, hexane was used to make up the volume. Next, equal volumes of the standard and sample were separately injected into the HPLC; the chromatogram was recorded and the responses were recorded for the major peaks. Finally, the percentage of tocotrienols/tocopherols in the sample was calculated. The HPLC conditions were: Column, Spherisorb Si 5 μm, 150 × 4.6 mm; mobile phase, hexane: ethyl acetate: acetic acid: 2,2-dimethoxypropane; ratio, 500:8.0:4.0:0.1; flow rate, 1 mL/min; wavelength, 295–300 nm; injection volume, 20 μm; mode, isocratic; temperature, 30 °C. Carotene was melted and thoroughly homogenized at a temperature of 60 °C to 70 °C and 0.1 g of oil sample was weighed into a 25 mL volumetric flask. The test oil sample was dissolved with a few milliliters of solvent and diluted to the scale and the oil solution and solvent were transferred to a 10 mm cuvette, respectively, and the absorbance was read at a wavelength of 446 nm using a spectrophotometer.

### 2.4. Histopathological Analysis

The experimental procedures of Masson’s trichrome staining and hematoxylin-eosin (HE) staining were conducted according to the manufacturer’s instructions. The pulmonary tissues were dehydrated in graded alcohol and fixed in paraffin blocks, whereafter they were cut into slices of 4 µm each. The slices were dewaxed with xylene, rehydrated with ethanol, and washed with distilled water. They were then kept with hematoxylin for 5 min, then with eosin for 2 min or the Masson’s compound dyeing solution for 5 min, and finally with bright green solution for 5 min. Conventional dehydration, tissue clearing, and sealing were performed. The scoring criteria were based on those previously reported [29].

### 2.5. Immunohistochemical (IHC) Determination of Collagen I

Tissue blocks smaller than 0.5 cm × 0.5 cm × 0.1 cm were taken for standby, then washed five times with phosphate-buffered saline (PBS). IHC was performed according to standard protocols. Briefly, after fixation, embedding, slicing, and dewaxing, antihelion was repaired with 0.01 M sodium citrate solution for 20 min and the endogenous peroxidase activity was quenched with 3% H_2_O_2_ for 10 min. The tissues were then left in 1% serum in PBS at 37 °C for 20 min to eliminate nonspecific binding. Following the blocking, the sections were rinsed and incubated overnight at 4 °C with collagen I antibody (AF7001, Affinity Biosciences, Beijing, China). Thereafter, biotinylated rabbit secondary antibody (SP-9001, ZSJQ-BIO, Guangdong, China) was added to the slices and left for 20 min at 37 °C. Horseradish enzyme, labeled streptomyces ovalbumin working solution (SA/HRP), was added to the slices at left at 37 °C for 20 min. The sections were developed by conventional dehydration, tissue clearing, diaminobenzidine (DAB) staining, and neutral resin sealing. Image-Pro Plus 6.0 software was used for assessing results and expressed as a percentage of the total analyzed areas.

### 2.6. Determination of Levels of Inflammatory Cytokines, Oxidative Stress and PF Markers

On day 28, the IL-1β, IL-6, MPO, TGF-β1, and TNFα levels in the lung tissues were analyzed using ELISA Kits. The standards supplied with the kits were used to generate the standard curve. The activities of GSH and CAT, as well as hydroxyproline (HYP), the contents of lipid peroxidation marker MDA, MPO, and MMP-7 in the rat lung tissues were measured using the ELISA kits, according to the manufacturer’s protocols.

### 2.7. Western Blotting

Western blot analysis was performed as described by Rong et al. [30]. The basic steps included protein concentration, polyacrylamide gel electrophoresis, film rotation, closing, primary antibody incubation, film washing, secondary antibody incubation, film washing, and enhanced chemiluminescence (ECL) development. The antibodies were: anti-TGF-β1 (ab215715, Abcam), anti-α-SMA (ab7817, Abcam), anti-Smad2 (ab40855, Abcam), anti-Smad3 (ab40854, Abcam), anti-Smad7 (25840-1-AP, Proteintech), anti-PI3K (ab191606, Abcam), anti-p-Akt^ser473^ (66444-1-Ig, Proteintech), anti-mTOR (ab134903, Abcam), anti-p-IkBα (ab133462, Abcam), anti-p-IKKβ (AF3010, Affinity), and anti-p-P65 (ab76302, Abcam).

### 2.8. qRT-PCR Analysis

qRT-PCR analysis was performed as described by Liu et al. [31]. The basic steps included the removal of residual genomic DNA, preparation of reverse transcription reaction system, reverse transcription program settings, and fluorescence quantitative PCR. Primer information is shown in Table 1.

### 2.9. Statistical Analysis

The data values were presented as the mean ± standard error of the mean (±SEM). The significance was assessed by one-way analysis of variance and Dunnett’s *t*-test was applied for post-multiple comparison, as appropriate; *p*-values of <0.05 were considered significant. Statistical significance was assessed using GraphPad Prism 9.0 software

## 3. Results

### 3.1. Compositions of TRF and Total Mixed-Carotene Complex 20% Oil Concentrate (Carotene)

The results obtained by high-performance liquid chromatography (HPLC) analyses and the spectrophotometer method of TRF and carotene are listed separately in Table 2 and Table 3, respectively, in which the content of the carotene mixture was found to exceed 20%, with β-carotene content being the highest. It was also inferred from the results presented in Table 3 that the tocotrienol content was as high as 39.8%.

### 3.2. Effects of TRF and Carotene on Weight (g), Feed Intake (g) and Lung/Body Weight Ratio (%)

During the 28-day treatment period, in addition to some weight loss during the first 2–3 days of modeling, the body weight of rats rose steadily (Figure 1a). In the process of modeling, the feed intake of the rats also was not stable and did not seem to be affected by the bleomycin infusion, despite some individual abnormal values (Figure 1b). The lungs were isolated and weighed immediately after the sacrifice, and the lung indices were subsequently calculated to evaluate the extent of deposited fibrosis. The lung/body weight ratio is an important basic index of pulmonary fibrosis; however, in the results of this modeling (Figure 1c), it was found that there was no significant difference between the groups.

### 3.3. Effects of TRF and Carotene on Histological Evaluation

Inflammatory situations were identified via HE staining and the deposition of collagen was examined via Masson’s trichrome staining in different sections. To assess the severity of lung injury, a quantitative scoring system was used to evaluate the extent of pulmonary parenchymal fibrosis (as noted in Section 2). Through the representative pictures of HE staining (Figure 2a), under the original magnifications of 100 and 400, the normal alveoli were evenly distributed and the epithelial cells were arranged in order with a clear boundary. In the model group, the alveoli were found to be obviously dilated, the pulmonary interstitia were bleeding extensively and the alveolar septa were narrowed and broken. The severity of the other groups was found to range between that of the control group and that of the model group, which was consistent with the quantitative analysis of pulmonary grades (lung inflammation) (Figure 2c). Thus, it was inferred that the carotene + TRF group had the least severe damage, and the treatment of carotene and TRF significantly reduced the score compared to the BLM group. Through Masson’s trichrome staining (Figure 2b), it was also revealed that the carotene + TRF group also had the highest efficacy in restoring morphological and functional status compared with other groups, whose overall situation was similar to that of HE staining. This concurred with the quantitative analysis of the collagen, which produced the same results. Compared with the model group, only the high-dose TRF group and the carotene and TRF group showed statistical differences (*p* < 0.05) (Figure 2c,d).

### 3.4. Immunohistochemical Determination of Collagen I

The increase in the collagen I ratio is one of the markers of PF development [32]. In this study, after 4-week exposure to BLM, the collagen I expression level was detected in lung tissue via IHC to investigate the effects of TRF and carotene on PF. In our experiments, 5 µm thick sections were stained to identify collagen fibers. Under meticulous observation, it was ascertained that the collagen I expression level was significantly up-regulated in the BLM group compared to the control group (Figure 3c), which was consistent with images obtained via IHC staining (Figure 3a,b). The images of magnification ×400 were more visible than those of magnification ×100. Thus, it was evident that the administrations of TRF and carotene obviously down-regulated the expression since there were significant differences between all groups and the model group, with the exception of the low-dose TRF group. Similarly, the results of the histological evaluation showed that the TRF and carotene group had the best preventive effect.

### 3.5. Effects of TRF and Carotene on Inflammatory Markers and Antioxidant Enzymes

The inflammatory activities of IL-1β, IL-6, MPO, TGF-β1, and TNF-α in the lung tissues of both normal control rats and BLM-induced lung fibrosis rat models were determined (Figure 4a–e). As shown, the proinflammatory cytokine influx was significantly increased by the administration of BLM. Specifically, BLM induced significant increases in the IL-1β level (from 12.17 ± 0.84 to 26.95 ± 3.46 mg/g protein; *p* < 0.05), the IL-6 level (from 71.78 ± 8.57 to 162.90 ± 19.53 mg/g protein; *p* < 0.05), MPO level (from 74.52 ± 1.37 to 155.80 ± 16.87 mg/g protein; *p* < 0.05), TGF-β1 level (from 116.4 ± 10.48 to 222.20 ± 11.63 mg/g protein; *p* < 0.05) and the TNF-α level (from 112.5 ± 4.30 to 249.90 ± 32.92 mg/g protein; *p* < 0.05) in comparison to the control group. However, TRF and carotene both showed good anti-inflammatory properties, as can be seen in Figure 4. Specifically, TRF and carotene obviously inhibited the increase in inflammatory markers, and the TRF and carotene group always had the best preventive effects.

The antioxidant levels of CAT, GSH, MDA, NO, and SOD in the lung tissue of normal control rats and BLM-induced lung fibrosis rat model were also determined (Figure 4f–j). As with the proinflammatory cytokine influx, the levels of antioxidant enzymes were significantly changed by the administration of BLM and the TRF and carotene group always showed the best preventive effects. For instance, MDA activity was reduced from 63.84 ± 6.39 to 24.81 ± 1.55 nmol/mg protein (*p* < 0.05) and NO activity was reduced from 29.94 ± 2.05 to 15.33 ± 1.62 nmol/mg protein (*p* < 0.05) compared with the model group and the TRF and carotene group. However, the preventive effect was not always dose dependent. For example, the high-dose TRF group was found to have the best preventive effect on CAT, MDA, NO, and SOD among the three doses, while the medium-dose TRF group was more effective than that on the GSH level.

### 3.6. Effects of TRF and Carotene on HYP and MMP-7 Levels

The collagen deposition (MMP-7 and HYP) in lung tissues was induced by the administration of BLM, as shown in Figure 5. Compared with the MMP-7 levels and HYP content in the control group, those in the BLM-induced lung fibrosis rats in the model group were found to have increased significantly by 109.44% (from 18.53 ± 0.89 to 38.81 ± 3.59 mg/g protein, *p* < 0.05) and 257% (from 0.59 ± 0.06 to 2.10 ± 0.40 mg/g wet tissue, *p* < 0.05), respectively. However, it was evident that collagen deposition had been prevented by the treatment of TRF and carotene, with the most effective preventive effect seen in the combined treatment group. Specifically, MMP-7 levels and HYP content were reduced from 38.81 ± 3.59 to 23.98 ± 4.90 mg/g protein (*p* < 0.05) and 2.10 ± 0.40 to 0.79 ± 0.05 mg/g wet tissue (*p* < 0.05).

### 3.7. TRF and Carotene Prevent PF by Suppressing TGF-β/Smad Signaling Pathway

TGF-β signal activation can render macrophages chemotactic in the lesion site, induce the proliferation and activation of fibroblasts, increase the synthesis of collagen, stimulate the expression of a large number of proinflammatory factors and fibrotic cytokines, and further enhance and sustain the fibrotic response [33]. In this study, the expressions of TGF-β1 protein, Smad2 protein, Smad3 protein, and α-SMA protein were significantly (*p* < 0.05) downregulated compared to those in the normal control rats, while Smad7 protein was upregulated. These results are consistent with previous reports that Smad7 can inhibit the TGF-β signal pathway [34,35,36]. TRF treatment (50, 100, and 200 mg/kg body wt/day) significantly and dose-dependently downregulated or upregulated TGF-β1, Smad2, Smad3, α-SMA, and Smad7 expressions (Figure 6a–f). The inhibition of proteins was most obvious in the combined intervention group (carotene + TRF group), while the intervention effect in the carotene group was consistent with that observed in the low-dose TRF group. The results of the reverse transcription-quantitative polymerase chain reaction (RT-qPCR) (Figure 6g–j) showed that the expressions of TGF-β1 mRNA, Smad2 mRNA, and Smad3 mRNA in the BLM-induced rats increased significantly compared to those expressions in the normal group, while they were downregulated obviously by TRF and carotene. Consistent with the results of the protein expression, the gene expression of Samd7 was also contrary to others. Carotene and TRF were found to inhibit the expressions of TGF-β1, α-SMA, Smad2, and Smad3 in the lung tissues of the BLM-induced rats, but increased the expression of Smad7 protein, indicating that the wonderful anti-pulmonary fibrosis mechanisms of carotene and TRF may be related to the TGF-β/Smad signal pathway.

### 3.8. TRF and Carotene Inhibit Pulmonary Fibrosis by Suppressing PI3K/Akt/mTOR Signaling Pathway

The PI3K/Akt/mTOR signaling pathway is an important intracellular signal transduction pathway related to cell growth, proliferation, survival, apoptosis, migration, protein synthesis, autophagy, and reverse transcription [37,38,39]. In addition, the inhibition of the PI3K/Akt/mTOR pathway can promote autophagy and improve pulmonary fibrosis [31,40]. Here, the expressions of PI3K protein, p-Akt^Ser473^ protein, and mTOR protein were significantly (*p* < 0.05) downregulated compared with those in the normal control rats. TRF treatment (50, 100, and 200 mg/kg body wt/day) significantly and dose-dependently downregulated PI3K, p-Akt^Ser473^, and mTOR expressions (Figure 7a–d). Our results showed that, compared with the control group, the phosphorylation of the PI3K/Akt/mTOR pathway decreased, most significantly in the combined intervention group. The results of RT-qPCR (Figure 7e–g) showed that the expressions of PI3KmRNA, AktmRNA, and mTORmRNA in the BLM-induced rats increased significantly compared to those of the normal group, while their expression levels were downregulated obviously by TRF and carotene. Specifically, the protein results were confirmed at the genetic level. All of the above findings thus suggest that TRF and carotene can inhibit the phosphorylation of the PI3K/Akt/mTOR signaling pathway.

### 3.9. TRF and Carotene Inhibit Pulmonary Fibrosis by Suppressing NF-κB Signaling Pathway

NF-κB is widely present in inflammatory cells. When stimulated, NF-κB is activated and participates in a variety of physiological and pathological processes [41]. The expression and activation of NF-κB and the expression level of inflammatory cytokines downstream of the NF-κB signaling pathway are positively correlated with the degree of the inflammatory response [42]. Our results showed that, compared with the control group, the phosphorylation of the NF-κB signaling pathway decreased, most significantly in the combined intervention group. The expressions of p-p65 protein, p-IkBα protein, and Ikkβ protein were significantly (*p* < 0.05) downregulated compared with those in normal control rats. TRF treatment (50, 100, and 200 mg/kg body wt/day) significantly and dose-dependently downregulated p-p65, p-IkBα, and Ikkβ expressions (Figure 8a–d). The results of RT-qPCR (Figure 8e–j) showed that the expressions of TNF-α mRNA, IFN-γ mRNA, IL-13 mRNA, NF-κB mRNA, IkBα mRNA, and Ikkβ mRNA in the BLM-induced rats increased significantly compared to those in the normal group, while their expression levels were downregulated obviously by TRF and carotene. It is worth noting that although TRF and carotene could significantly downregulate the protein expression of NF-κB in lung tissue and inhibit the activation of NF-κB, the gene expression of NF-κB did not show a dose-dependent relationship in the low-, medium-, or high-dose TRF groups. In summary, TRF and carotene can inhibit the activation of the NF-κB signaling pathway and the release of downstream inflammatory factors to reduce pulmonary inflammation and improve pulmonary fibrosis.

## 4. Discussion

While there are many causes of PF, their pathogeneses are similar. In the early stage, pulmonary inflammation is the main cause, followed by alveolar injury, and then chronic inflammation and tissue repair. Finally, the excessive proliferation of fibroblasts and abnormal repair of lung tissue lead to EMT, and the resulting deposition of a large number of collagen fibers in the lung stroma to form PF, which is, thus, the common outcome of many lung diseases [43,44]. At present, two drugs have been approved by the United States Food and Drug Administration (FDA) for the treatment of idiopathic PF, namely pirfenidone and nintedanib. Although these two drugs can delay the deterioration of pulmonary function in patients with PF, their clinical application is limited due to adverse reactions such as rash, nausea, diarrhea, and liver function injury [45]. Moreover, these medications are very costly and, consequently, may be unobtainable for most patients. There, thus, is an urgent need for a more appropriate method of resisting PF, and, indeed, the focus on prevention is a more cost-effective approach than treatment. Consequently, this paper aimed to study potential interventions for the prevention of PF.

The growth rate of rat body weight can reflect the trend of organ and tissue cell proliferation, as well as the physiological and biochemical indexes at different stages [46]. During the development of PF, the lung mass increases due to inflammation, exudation, cell swelling, and capillary congestion. Therefore, the lung coefficient is regarded as one of the indicators of the degree of PF and of pulmonary edema [47]. The pathological results (Figure 2) in this study showed that our model was successful; however, the record of weight values (Figure 1a) was different from the results of previously reported PF experiments, with the weight of their experimental rats decreasing significantly and for a long time after modeling [29,32]. Interestingly, no significant difference between groups was found in the lung index, possibly because our gentle modeling method did not cause continuous weight loss in the rats or malignant damage to their lungs. Inflammation and fibrosis are two important aspects of the pathogenesis of PF and have, thus, become potential targets in the treatment thereof [48]. While the anti-inflammatory effects of vitamins A and E are well known, updated research has shown that tocotrienols possess superior antioxidant and anti-inflammatory properties to those of α-tocopherol [25], and a multitude of studies examining the role of vitamin A and carotenoids as biological antioxidants have been published over the last 15 years [47,49,50]. It is worth noting that there are few studies on the relationship between vitamins and PF, while even fewer related to vitamins A and E have focused on the changes in inflammation and heavy metal contents in BLM-induced rats, and none have reported the study of the mechanisms therein [51,52,53]. Consequently, the purpose of this paper was to study the potential preventive effects of TRF and carotene on BLM-induced rats and to explore the possible pathways involved therein.

Studies show that relevant natural products can prevent and treat PF by inhibiting inflammation, ameliorating oxidative stress, and regulating EMT [12], with the mechanisms involving TGF-β1/Smad, p38 MAPK, PI3K/Akt, Nrf2-Nox4, NF-κB, and AMPK signaling pathways. In this study, three pathways were selected for preliminary research based on the characteristics of TRF and carotene, namely TGF-β/Smad, PI3K/AKT/mTOR, and NF-κB signaling pathways. It was found that TRF and carotene significantly prevented pathological changes in the BLM-induced rats, inhibited the inflammatory response of lungs, reduced the content of MDA in lung tissue, increased the activity of SOD in lung tissue, and inhibited oxidative stress in a certain dose-dependent manner (Figure 4). The preventive effect of the combined group (TRF + carotene) was generally the best, and the effect of the carotene group was always similar to that of the low-dose TRF group. The results of this experiment showed that TRF and carotene significantly inhibited the formation of BLM-induced PF in rats, mainly by inhibiting inflammatory response, oxidative stress, and fibrosis. Furthermore, it was found that the possible mechanism involved the inhibition of the inflammatory response by down-regulating the expression of NF-κB protein and inhibiting the release of inflammatory downstream cytokines TNF-α, IFN-γ, and IL-13 of the NF-κB signal pathway, inhibiting the TGF-β/Smad signaling pathway by down-regulating the protein expressions of TGF-β1, Smad2/3, and collagen Ⅰ in lung tissue, and inhibiting lung EMT by up-regulating the interstitial cell marker α-SMA. TRF and carotene also protected against PF via the inhibition of the PI3K/AKT/mTOR signaling pathway by down-regulating the expressions of PI3K protein, p-AKT protein, and mTOR protein. It is, thus, evident that the roles of TRF and carotene were reflected in these three pathways. Based on the above results, we have reason to believe that TRF and carotene have good potential for development into anti-fibrosis preventive products to provide protection for people working in adverse working environments and, thus, at risk of developing PF.

## 5. Conclusions

In summary, we showed the potential of TRF and carotene in the restoration of the antioxidant system and the inhibition of inflammatory cytokines (IL-1β, IL-6, MPO, TGF-β1, and TNF-α), oxidative stress (MDA), and extracellular matrix (HYP and MMP-7). Histopathological findings revealed that TRF and carotene treatments significantly ameliorated BLM-induced lung injury. This is the first report demonstrating that TRF and carotene have a certain preventive effect on BLM-induced PF in rats and that the possible mechanism therein involves the TGF-β/Smad, PI3K/Akt/mTOR, and NF-κB signaling pathways.

## Figures and Tables

**Figure 1 nutrients-14-01094-f001:**
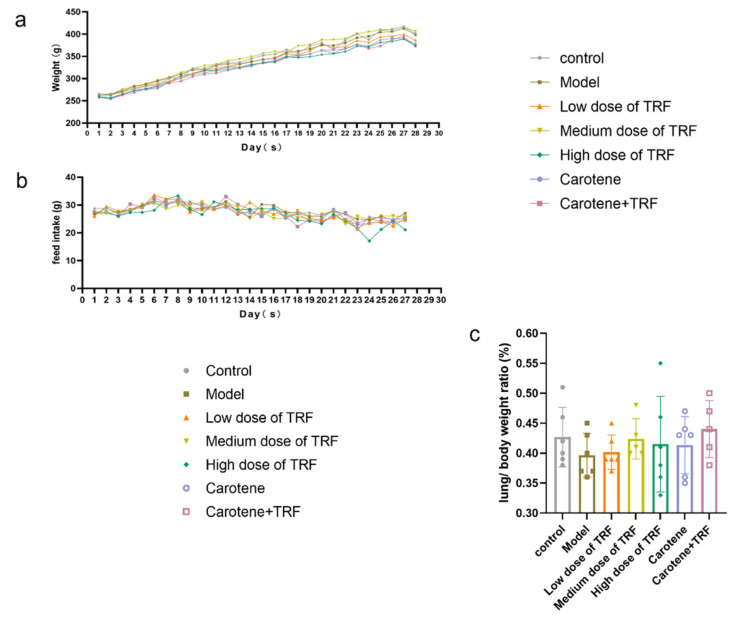
Effects of TRF and carotene on weight (g), feed intake (g) and lung/body weight ratio (%): (**a**) Body weight changes in rat groups recorded every other day after BLM administration; (**b**) feed intake changes in rat groups recorded every other day after BLM administration; (**c**) lung/body weight ratios in rat groups on final day. Data are expressed as the mean ± SD; *n* = 6. There was no significant difference between the groups.

**Figure 2 nutrients-14-01094-f002:**
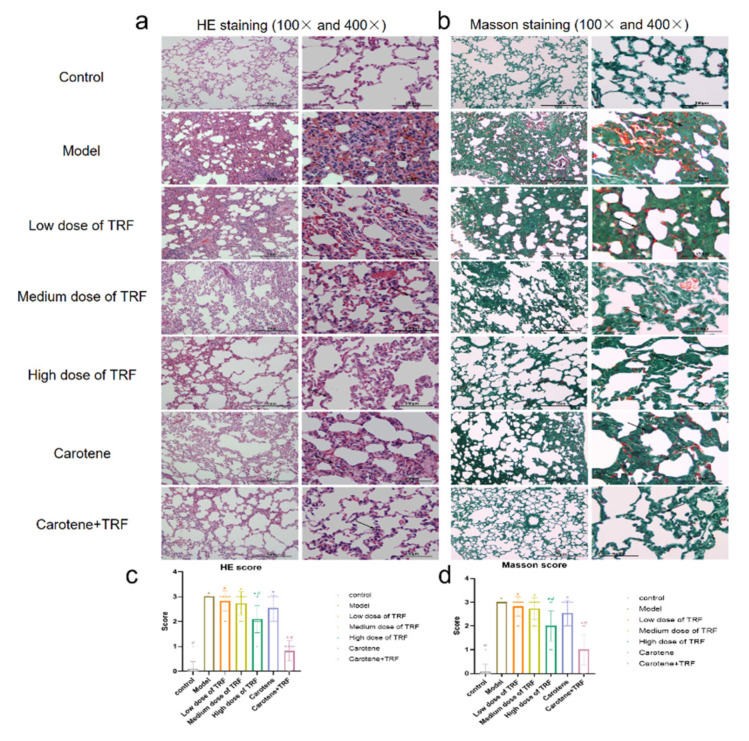
TRF and carotene inhibited BLM-induced PF. (**a**) Representative HE staining of the lungs, with original magnification ×100 and ×400; (**b**) representative Masson’s trichrome staining of the lungs, with original magnification ×100 and ×400; (**c**) quantitative analysis of lung inflammation, *n* = 11; (**d**) quantitative analysis of the collagen, *n* = 11. Note: compared with CON group, # represents *p* < 0.05; compared with MOD group, * represents *p* < 0.05.

**Figure 3 nutrients-14-01094-f003:**
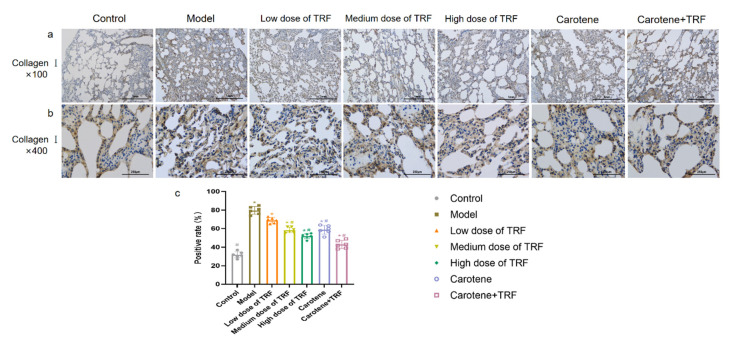
TRF and carotene ameliorated bleomycin-induced pulmonary fibrosis. (**a**) Representative images of IHC staining for collagen I in the lungs, with original magnification ×100; (**b**) representative images of IHC staining for collagen I in the lungs, with original magnification ×400; (**c**) quantitative analysis of the lung inflammation, *n* = 6. Note: compared with the CON group, # represents *p* < 0.05; compared with MOD group, * represents *p* < 0.05.

**Figure 4 nutrients-14-01094-f004:**
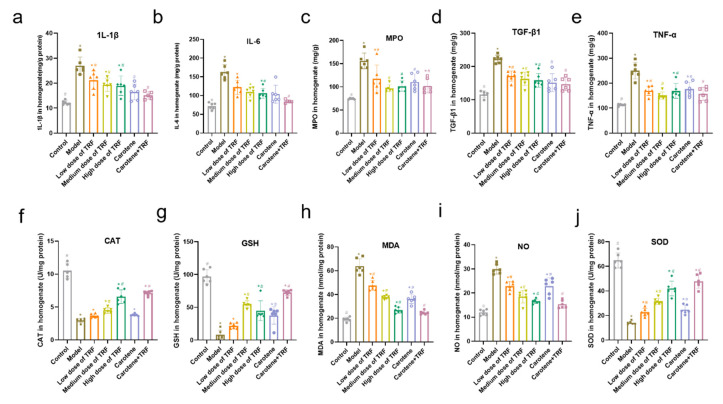
Effects of TRF and carotene on inflammatory markers and antioxidant enzymes: (**a**–**e**) Effects on pro-inflammatory cytokine levels in the lung tissues of BLM-induced lung fibrosis, *n* = 6; (**f**–**j**) effects on antioxidant enzymes and oxidative stress markers in the lung tissues of BLM-induced lung fibrosis, *n* = 6. Note: compared with CON group, # represents *p* < 0.05; compared with MOD group, * represents *p* < 0.05.

**Figure 5 nutrients-14-01094-f005:**
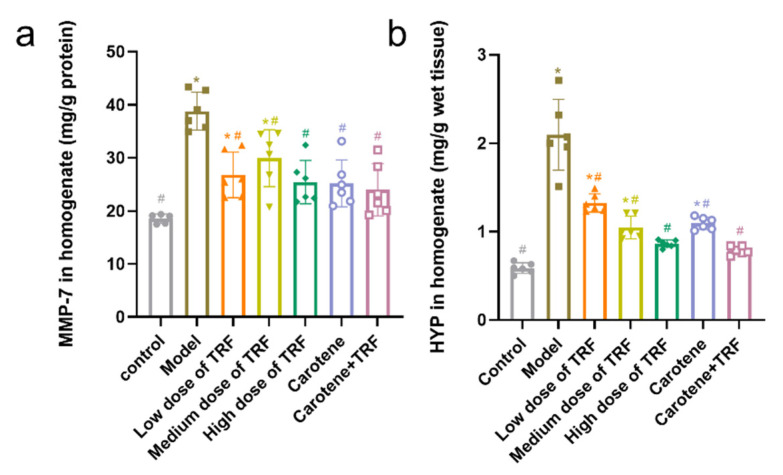
Effects of TRF and carotene on HYP and MMP-7 levels in lung tissue: (**a**), Effects of TRF and caroteneon on MMP-7 in the lung tissues of BLM-induced lung fibrosis, *n* = 6; (**b**), Effects of TRF and caroteneon on HYP in the lung tissues of BLM-induced lung fibrosis, *n* = 6; Note: compared with CON group, # represents *p* < 0.05; compared with MOD group, * represents *p* < 0.05.

**Figure 6 nutrients-14-01094-f006:**
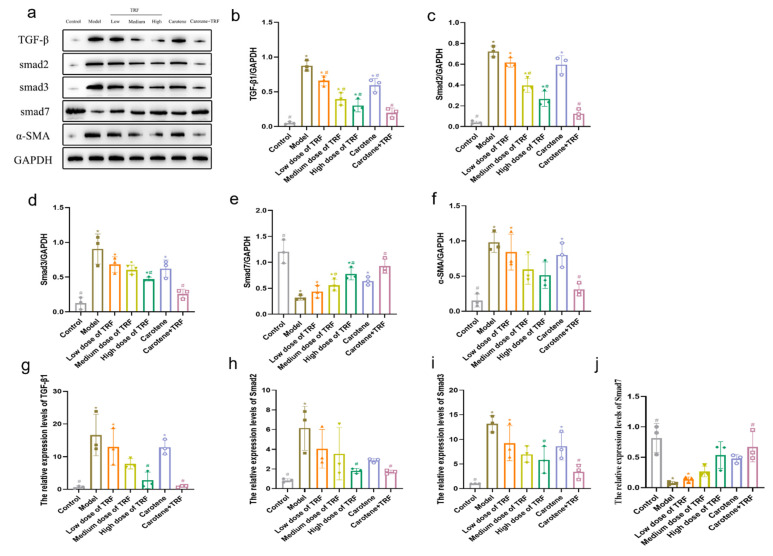
TRF and carotene protected against pulmonary fibrosis via the inhibition of the TGF-β/Smad signaling pathway: (**a**–**f**) TRF and carotene decreased the expression of fibrosis proteins induced by BLM in SD rats. The protein expressions of (**b**) TGF-β1, (**c**) Smad2, (**d**) Smad3, € Smad7, (**f**), and (**g**) α-SMA in the lung samples were examined by Western blotting analysis; *n* = 3; (**g–j**) representative statistical analyses of (**g**) TGF-β1, (**h**) Smad2, (**i**) Smad3, and (**j**) Smad7 mRNA by RT-qPCR; *n* = 3. Note: compared with CON group, # represents *p* < 0.05; compared with MOD group, * represents *p* < 0.05.

**Figure 7 nutrients-14-01094-f007:**
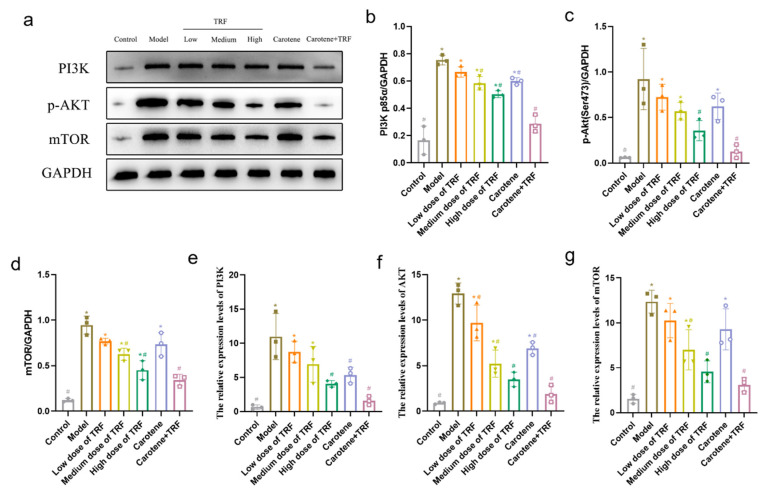
TRF and carotene protected against pulmonary fibrosis via the inhibition of the PI3K/AKT/mTOR signaling pathway: (**a**–**d**) TRF and carotene decreased the expression of fibrosis proteins induced by BLM in SD rats. The protein expressions of (**b**) PI3K, (**c**) p-AKT, and (**d**) mTOR in the lung samples were examined by Western blotting analysis, *n* = 3: (**e**–**g**) representative statistical analyses of (**e**) PI3K, (**f**) AKT, and (**g**) mTOR mRNA by RT-qPCR, *n* = 3. Note: compared with CON group, # represents *p* < 0.05; compared with MOD group, * represents *p* < 0.05.

**Figure 8 nutrients-14-01094-f008:**
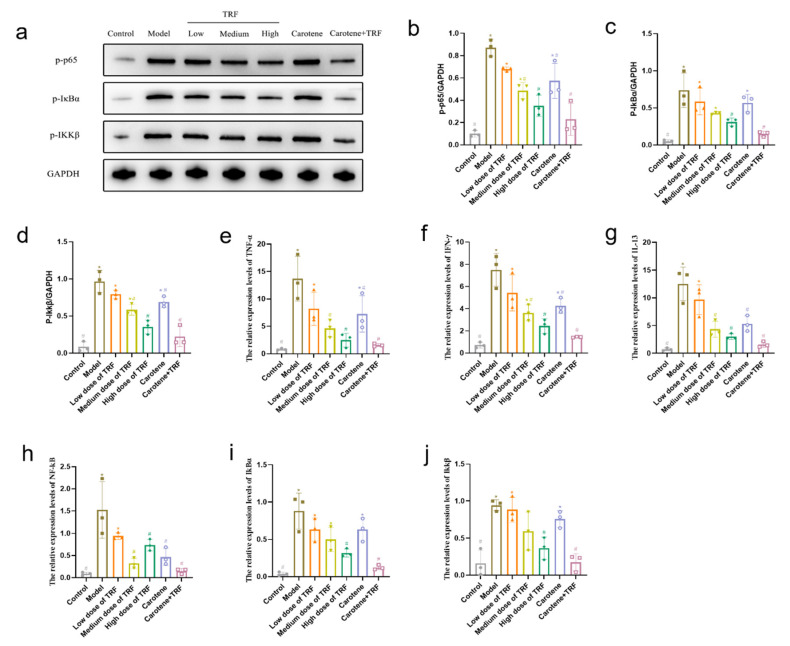
TRF and carotene protected against pulmonary fibrosis via the inhibition of the NF-κB signaling pathway: (**a**–**d**) TRF and carotene decreased the expression of fibrosis proteins induced by BLM in SD rats. The protein expressions of (**b**) p-p65, (**c**) p-IkBα, and (**d**) Ikkβ in lung samples were examined by Western blotting analysis, *n* = 3; (**e**–**j**) representative statistical analyses of (**e**) TNF-α, (**f**) IFN-γ, (**g**) IL-13, (**h**) NF-κB, (**i**) IkBα, and (**j**) Ikkβ mRNA by RT-qPCR, *n* = 3. Note: compared with CON group, # represents *p* < 0.05; compared with MOD group, * represents *p* < 0.05.

**Table 1 nutrients-14-01094-t001:** Primer information.

mRNA	Forward Primers	Reverse Primers
TGF-β1	TCGCCCTTTCATTTCAGAT	TTTGCCGATGCTTTCTTG
Smad2	AGGTGTCTCATCGGAAAG	CTCTGGTAGTGGTAAGGGT
Smad3	AGCTTACAAGGCGGCACA	TGGGAGACTGGACGAAAA
Smad7	CTTCCTCCGATGAAACCG	TCGAGTCTTCTCCTCCCAGTA
PI3K	GAAACCCAGTCACCTAGGGC	GGTGGGCAGTACGAACTCAA
AKT	GAGGAGCGGGAAGAGTG	GTGCCCTTGCCCAGTAG
mTOR	GGTGGACGAGCTCTTTGTC	AGGAGCCCTAACACTCGGAT
TNF-α	TGAGCACAGAAAGCATGATC	CATCTGCTGGTACCACCAGTT
IFN-γ	TTGCAGCTCTGCCTCAT	TTCGTGTTACCGTCCTT
IL-13	CTCGCTTGCCTTGGTGG	TGATGTTGCTCAGCTCCTC
NF-κB	CTGTTTCCCCTCATCTTTCC	GTGCGTCTTAGTGGTATCTGTG
IkBα	CCAACTACAACGGCCACA	CAACAGGAGCGAGACCAG
Ikkβ	CATTGTTGTTAGCGAGGAC	CCCTTTGCCGAGGTTGC
GAPDH	AAGAAGG TGGTGAAGCAGGC	TCCACCACCCT GTTGCTGTA

Note: TGF-β1, transforming growth factor beta 1; Smad2, Smad2; Smad3, Smad3; Smad7, Smad7; PI3K, phosphatidylinositol 3-kinase; AKT, (protein kinase B, PKB); mTOR, mammalian target of rapamycin; TNF-α, tumor necrosis factor-alpha; IFN-γ, interferon γ; IL-13, interleukin-13; NF-κB, nuclear factor kappa-B; IkBα, NF-kappa-B inhibitor alpha; Ikkβ, inhibitor of nuclear factor kappa-B kinase.

**Table 2 nutrients-14-01094-t002:** Compositions of total mixed-carotene complex 20% oil concentrate (carotene).

Compositions	Values (mg/g)
α-Carotene	65 ± 1.15
β-Carotene	135 ± 2.31
γ-Carotene	0.5 ± 0.06
Lycopene	0.1 ± 0.01
Total mixed-carotene complex	200.6

**Table 3 nutrients-14-01094-t003:** TRF compositions.

Compositions	Value (wt/wt)
α-Tocopherol	12.5 ± 0.17
α-Tocotrienol	12.8 ± 0.06
β-Tocotrienol	2.0 ± 0.12
γ-Tocotrienol	19.5 ± 0.35
δ-Tocotrienol	5.5 ± 0.06
Total mixed tocotrienols	39.8
Tocotrienol/tocopherol complex	52.3

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
