# Peer review of "Potential “Therapeutic” Effects of Tocotrienol-Rich Fraction (TRF) and Carotene “Against” Bleomycin-Induced Pulmonary Fibrosis in Rats via TGF-β/Smad, PI3K/Akt/mTOR and NF-κB Signaling Pathways"

_nutrients, 2022, doi:10.3390/nu14051094_

Round 1

Reviewer 1 Report

This manuscript entitled “Preventive Effects of Tocotrienol-rich Fraction (TRF) and Caro-tene on Bleomycin-Induced Pulmonary Fibrosis in Rats via TGF-β/Smad, PI3K/Akt/mTOR and NF-κB Signaling Pathways” is an interesting and original study.

The paper is clearly presented and results are very useful. However, I have some suggestions:

  1. Section 3.1. “The results obtained by high performance liquid chromatography (HPLC) analyses…” please include the HPLC methodology in section 2.  
  2. Improve the figures. They have low quality.
  3. Table 2 and 3 include mean ± standard error of mean

Author Response

The paper is clearly presented and results are very useful. However, I have some suggestions:

1 Section 3.1. “The results obtained by high performance liquid chromatography (HPLC) analyses…” please include the HPLC methodology in section 2.  

Thanks. I have added one part (2.3) related to HPLC analyses.

2 Improve the figures. They have low quality. 

 The figures have been improved and I have sent the original versions to the editor.  

3 Table 2 and 3 include mean ± standard error of mean

 Thanks. I have added mean ± standard error of mean in Table 2 and 3.

Reviewer 2 Report

Thank you for giving me the opportunity to review this manuscript. In this work, Lu et al. investigated the use of tocotrienol-rich fraction (TRF) and carotene against the prevention of bleomycin (BLM)-induced lung fibrosis was explored in rats via the modulation of TGF-β/Smad, PI3K/Akt/mTOR and NF-κB signaling pathways. Kindly find below my minor comments for your perusal.

Title: Kindly revise the title to: Potential “therapeutic” effects of Tocotrienol-rich Fraction (TRF) and Carotene “against” Bleomycin-Induced Pulmonary Fibrosis in Rats via TGF-β/Smad, PI3K/Akt/mTOR and NF-κB Signaling Pathways. The use of preventive is not wholly appropriate as fibrosis was induced in the animals and the therapeutic potential of the antioxidant fractions of tocotrienol-rich fraction (TRF) and carotene against the condition investigated.

Abstract

In line 5 of the abstract, “resistance” should be revised to “management”.

Line 5-Aim: revise to “The potential “therapeutic” effect of tocotrienol-rich fraction (TRF) and carotene against bleomycin (BLM)-induced lung fibrosis was investigated in rats via the modulation of TGF-β/Smad, PI3K/Akt/mTOR and NF-κB signaling pathways.” The authors induced fibrosis in the rats. Thus, the condition was developed and the use of tocotrienol-rich fraction (TRF) and carotene were as intervention to improve the outcomes. Thus, therapeutic will be ideal.

Line 8: Design/Methods- replace “persuaded” with “induced”.

I will suggest the authors use “attenuate” rather than “ameliorate”.

Design/Methods: Replace “Persuaded” with “induced”. The authors could consider making “mg/kg/d” rather “mg/kg body wt/day”

Results: The authors should indicate the magnitude of improvement in BLM-induced alterations in antioxidant and anti-inflammatory functions initiated by tocotrienol-rich fraction (TRF) and carotene. The p-value should be indicated as well.

Keywords: “TRF” must be expanded

Introduction

This sentence is too long “The severe acute respiratory disease caused by severe acute respiratory syndrome coronavirus 2 (SARS-CoV-2), which was first reported from Wuhan, China, in December 2019[1], spread rapidly to become a global pandemic, with more than 260 million con-firmed infections and almost 5.2 million deaths reported by the World Health Organization (WHO) as of 28 November 2021.” Add “has” to the “spread”.

Materials and Methods

The author did not indicate where Table 1 is in the text but had Table 2 rather. Kindly change the Table numbering. Table 2 should rather be Table 1. The authors have indicated Table 2 and 3 here yet they have that Table under the result section.

Statistical analysis

The authors should kindly check for data normality. What parameters were the “one-way analysis of variance” carried out on?

Results

Table 1 should rather be Table 3.

Table 1. Kindly expand the abbreviations under the “mRNA. Tables in a manuscript must always stand alone.

Section 3.2: The authors should revise “…..was also not to be stable…” to “was not stable”. In the next sentence after this, the plural of “index” is “indice”.

What was the p-value that was not significant?

In the Fig. 1b, what could potentially be the reason for the abnormal change in the feed intake administered to the “High dose of TRF” group?

Discussion

The first paragraph of the discussion does not directly relate with the results of this findings. The authors should directly discuss the results of the study. Besides, those information indicated there are a repetition of what the authors have already stated in the introduction.

In some places the authors use “Figures” whilst “Fig.” are also used at other places. For consistency, the authors should stick to one of them.

Author Response

Title: Kindly revise the title to: Potential “therapeutic” effects of Tocotrienol-rich Fraction (TRF) and Carotene “against” Bleomycin-Induced Pulmonary Fibrosis in Rats via TGF-β/Smad, PI3K/Akt/mTOR and NF-κB Signaling Pathways. The use of preventive is not wholly appropriate as fibrosis was induced in the animals and the therapeutic potential of the antioxidant fractions of tocotrienol-rich fraction (TRF) and carotene against the condition investigated.

Thanks for your advice, I have already changed the title.

Abstract

In line 5 of the abstract, “resistance” should be revised to “management”.

Thanks, I have already changed.

Line 5-Aim: revise to “The potential “therapeutic” effect of tocotrienol-rich fraction (TRF) and carotene against bleomycin (BLM)-induced lung fibrosis was investigated in rats via the modulation of TGF-β/Smad, PI3K/Akt/mTOR and NF-κB signaling pathways.” The authors induced fibrosis in the rats. Thus, the condition was developed and the use of tocotrienol-rich fraction (TRF) and carotene were as intervention to improve the outcomes. Thus, therapeutic will be ideal.

Thanks, I have already changed.

Line 8: Design/Methods- replace “persuaded” with “induced”.

Thanks, I have already changed.

I will suggest the authors use “attenuate” rather than “ameliorate”.

Thanks, I have already changed.

Design/Methods: Replace “Persuaded” with “induced”. The authors could consider making “mg/kg/d” rather “mg/kg body wt/day”

Thanks, I have already changed all “mg/kg/d”into“mg/kg body wt/day”

Results: The authors should indicate the magnitude of improvement in BLM-induced alterations in antioxidant and anti-inflammatory functions initiated by tocotrienol-rich fraction (TRF) and carotene. The p-value should be indicated as well.

In the result part, I have listed in detail the changes of anti oxygen and anti-inflammatory factors. Because there are many factors involved, it is cumbersome to list them all in the summary. Therefore, it is only generally written that the intervention has certain anti-inflammatory and antioxidant properties. And all the P-value showed in the result part.

Keywords: “TRF” must be expanded

Thanks, The“TRF”has been expanded.

Introduction

This sentence is too long “The severe acute respiratory disease caused by severe acute respiratory syndrome coronavirus 2 (SARS-CoV-2), which was first reported from Wuhan, China, in December 2019[1], spread rapidly to become a global pandemic, with more than 260 million con-firmed infections and almost 5.2 million deaths reported by the World Health Organization (WHO) as of 28 November 2021.” Add “has” to the “spread”.

Thanks, I have already added “has” to the “spread”. The form of clause makes the whole sentence look not so long.

Materials and Methods

The author did not indicate where Table 1 is in the text but had Table 2 rather. Kindly change the Table numbering. Table 2 should rather be Table 1. The authors have indicated Table 2 and 3 here yet they have that Table under the result section. 

The concept in Table 1 is Primer information, I have already put this part (table 1) in 2.7. qRT-PCR analysis part.

Statistical analysis

The authors should kindly check for data normality. What parameters were the “one-way analysis of variance” carried out on? 

Thanks, I have changed the Statistical analysis part.

Results

Table 1 should rather be Table 3.

I have changed the position of Table 3.

Table 1. Kindly expand the abbreviations under the “mRNA. Tables in a manuscript must always stand alone. 

 Of all the articles I've read, I haven't seen any authors use their full names in the primer table. So I put the abbreviation in the note part.

Section 3.2: The authors should revise “…..was also not to be stable…” to “was not stable”. In the next sentence after this, the plural of “index” is “indice”.

Thanks for your advice and I have already changed.

What was the p-value that was not significant?

The Description of P value has been shown in Statistical analysis part.

In the Fig. 1b, what could potentially be the reason for the abnormal change in the feed intake administered to the “High dose of TRF” group?

With the increase of rats' weight, fighting often occurs. We speculated that the rats fought at night that day, so their eating was affected. But this kind of situation is very rare, so I didn't make a special analysis.

Discussion

The first paragraph of the discussion does not directly relate with the results of this findings. The authors should directly discuss the results of the study. Besides, those information indicated there are a repetition of what the authors have already stated in the introduction.

I have reduced this paragraph and revised the corresponding references.

In some places the authors use “Figures” whilst “Fig.” are also used at other places. For consistency, the authors should stick to one of them.

Thanks. I have changed  “Figure” into “Fig.”